# Revealing Subject-Specific Temporal Trajectories From Longitudinal Data

### Christos Chatzis
Dept. of Data Science and Knowledge
Discovery, Simula Metropolitan
Center for Digital Engineering
Oslo, Norway
Faculty of Technology, Art and
Design, Oslo Metropolitan University
Oslo, Norway
christos@simula.no

### David Horner
Copenhagen Prospective Studies on
Asthma in Childhood, Herlev and
Gentofte Hospital, University of
Copenhagen
Copenhagen, Denmark
david.horner@dbac.dk

### Rasmus Bro
Dept. of Food Science, University of
Copenhagen
Copenhagen, Denmark
rb@food.ku.dk

### Ann-Marie Malby Schoos
Copenhagen Prospective Studies on
Asthma in Childhood, Herlev and
Gentofte Hospital, University of
Copenhagen
Copenhagen, Denmark
Dept. of Pediatrics, Copenhagen
University Hospital – Næstved,
Slagelse and Ringsted
Slagelse, Denmark
Dept. of Pediatrics, Copenhagen
University Hospital – Amager and
Hvidovre
Hvidovre, Denmark
Dept. of Clinical Medicine, University
of Copenhagen
Copenhagen, Denmark
ann-marie.schoos@dbac.dk

### Morten A. Rasmussen
Copenhagen Prospective Studies on
Asthma in Childhood, Herlev and
Gentofte Hospital, University of
Copenhagen
Copenhagen, Denmark
Dept. of Food Science, University of
Copenhagen
Copenhagen, Denmark
morten.arendt@dbac.dk

### Evrim Acar
Dept. of Data Science and Knowledge
Discovery, Simula Metropolitan
Center for Digital Engineering
Oslo, Norway
evrim@simula.no

## Abstract

Temporal multivariate data is ubiquitous in many domains, collected over time at planned visits in longitudinal cohorts or at specific intervals in challenge tests. While the analysis of such data often focuses on revealing temporal patterns common across subjects, subject-specific differences in those patterns hold the promise to enhance our understanding of underlying mechanisms and facilitate personalized approaches. However, reliably extracting such temporal patterns from longitudinal multivariate data remains an open challenge. In this work, we demonstrate that coupled matrix factorizations (CMF) and PARAFAC2 are effective tools for capturing such subject-specific temporal profiles, focusing on two novel applications. In longitudinal metabolomics, CMF and PARAFAC2 reveal differences in postprandial metabolic responses of individuals according to body mass index (BMI) and insulin resistance (IR) measures. In sensitization data analysis, both methods capture differences in temporal trajectories of children according to delivery mode, insights that CANDECOMP/PARAFAC (CP), the current state of the art, fails to uncover. We also demonstrate the reproducibility and replicability of the extracted patterns.

## CCS Concepts

• **Information systems** → **Data mining**; • **Computing methodologies** → **Factorization methods**.

## Keywords

Temporal data, PARAFAC2, Coupled matrix factorization

## 1 Introduction

The analysis of temporal multiway data is a prevalent problem across many domains, from meal challenge tests in metabolomics [10, 20] to allergen sensitization measured over childhood [17, 19]. A common goal is to uncover the underlying patterns and their temporal trajectories and to understand why they may differ across entities, e.g. subjects. Tensor decompositions such as CANDECOMP/PARAFAC (CP) have been successfully applied to such data [11, 19, 20], extracting interpretable patterns. However, CP extracts a single shared temporal profile per component, revealing only scaling differences across subjects; individual shape differences in temporal trajectories remain hidden.

In this work, we demonstrate that coupled matrix factorizations (CMF) and PARAFAC2, more structurally flexible techniques that

**Figure 1: CMF/PARAFAC2 models revealing subject-specific time profiles from a *metabolites* by *time* by *subjects* tensor, and an *allergens* by *time* by *subjects* tensor.**

allow for slice-specific factor matrices, are effective tools for reliably capturing such subject-specific temporal shape differences from longitudinal data. For PARAFAC2, such a setting has been previously explored in neuroscience [12, 13], electronic health records [14] and microbiome analysis [6]. Here, we focus on two novel applications from the Copenhagen prospective studies on asthma in childhood 2000 (COPSAC$_{2000}$) cohort [2]: analysis of longitudinal metabolomics data from a postprandial meal challenge test, and allergen sensitization data collected over the first 18 years of life. In metabolomics, CMF and PARAFAC2 reveal differences in metabolic response according to body mass index (BMI) and insulin resistance (IR) measures. In sensitization analysis, they capture differences in temporal trajectories that can be stratified according to birth/delivery mode. We assess the reproducibility and replicability of extracted patterns [1].

## 2 Methods

Assuming the input of a third-order tensor $\mathcal{X} \in \mathbb{R}^{I \times J \times K}$, we now briefly define the tensor factorization techniques utilized.

*CP.* An $R$-component CP model [3, 7] factorizes $\mathcal{X}$ as a sum of $R$ rank-one tensors, i.e. $\mathcal{X} \approx \sum_{r=1}^{R} \mathbf{a}_r \circ \mathbf{b}_r \circ \mathbf{c}_r = [\![\mathbf{A}, \mathbf{B}, \mathbf{C}]\!]$, where $\mathbf{A} \in \mathbb{R}^{I \times R}$, $\mathbf{B} \in \mathbb{R}^{J \times R}$, $\mathbf{C} \in \mathbb{R}^{K \times R}$ and $\mathbf{D}_k = \text{diag}(\mathbf{c}_{k,:})$. We analyze $\mathcal{X}$ using a CP model with non-negativity constraints on all modes in this work by solving the following optimization problem:

$$\min_{\mathbf{A}, \mathbf{B}, \{\mathbf{D}_k\}_{k \leq K}} \sum_{k=1}^{K} \left\| \mathbf{X}_k - \mathbf{A}\mathbf{D}_k\mathbf{B}^\top \right\|_F^2, \tag{1}$$
$$\text{subject to} \quad \mathbf{A}, \mathbf{B}, \{\mathbf{D}_k\}_{k \leq K} \geq 0$$

*PARAFAC2.* The PARAFAC2 model [8] extends CP by introducing slice-specific factor matrices $\{\mathbf{B}_k\}_{k=1}^{K}$, with each $\mathbf{B}_k \in \mathbb{R}^{J \times R}$, while retaining $\mathbf{A} \in \mathbb{R}^{I \times R}$ and $\mathbf{D}_k = \text{diag}(\mathbf{c}_{k,:}) \in \mathbb{R}^{R \times R}$ as in CP. We use a PARAFAC2 model with non-negativity constraints in all

modes by solving the following optimization problem:

$$\min_{\mathbf{A}, \{\mathbf{D}_k, \mathbf{B}_k\}_{k \leq K}} \sum_{k=1}^{K} \left\| \mathbf{X}_k - \mathbf{A}\mathbf{D}_k\mathbf{B}_k^\top \right\|_F^2 \tag{2}$$
$$\text{subject to} \quad \mathbf{A}, \{\mathbf{B}_k, \mathbf{D}_k\}_{k \leq K} \geq 0, \{\mathbf{B}_k\}_{k \leq K} \in \mathcal{P}$$

where $\mathcal{P} = \{\{\mathbf{B}_k\} \mid \mathbf{B}_k^\top \mathbf{B}_k = \mathbf{\Phi}, \forall k\}$, also referred to as the PARAFAC2 constraint, ensures uniqueness up to trivial ambiguities.

*CMF.* CMF [18] couples the factorization of each frontal slice of $\mathcal{X}$ through a shared factor matrix $\mathbf{A} \in \mathbb{R}^{I \times R}$ and slice-specific matrices $\{\mathbf{B}_k\}_{k \leq K}$, with each $\mathbf{B}_k \in \mathbb{R}^{J \times R}$. Without additional constraints, the model is non-identifiable since $\mathbf{A}\mathbf{B}_k^\top = \mathbf{A}\mathbf{Q}^{-1}(\mathbf{Q}\mathbf{B}_k^\top) = \bar{\mathbf{A}}\bar{\mathbf{B}}_k^\top$ for any invertible $\mathbf{Q}$. Non-negative CMF, as utilized in this work can be estimated from:

$$\min_{\mathbf{A}, \{\mathbf{B}_k\}_{k \leq K}} \sum_{k=1}^{K} \left\| \mathbf{X}_k - \mathbf{A}\mathbf{B}_k^\top \right\|_F^2 \tag{3}$$
$$\text{subject to} \quad \mathbf{A}, \{\mathbf{B}_k\}_{k \leq K} \geq 0$$

*Uncovering temporal trajectories.* In order to extract individual temporal trajectories from the data with CP, PARAFAC2 or CMF, we need to appropriately re-order the data such that time is the second mode, features correspond to the first and subjects to the third. Applying (non-negative) CP to such a third-order tensor yields the following interpretation: $\mathbf{A}$, $\mathbf{B}$ and $\mathbf{C}$ reflect the feature importance, temporal profile and subject participation of each uncovered pattern. PARAFAC2's interpretation follows suit, but instead of a single temporal profile for each pattern, the slice-specific factor matrices $\{\mathbf{B}_k\}_{k=1}^{K}$, one per subject in this case, allow for the recovery of subject-specific temporal profiles (Figure 1). The interpretation is similar for CMF: $\mathbf{A}$ reflects feature importance while $\{\mathbf{B}_k\}_{k=1}^{K}$ recover individual trajectories for each subject; unlike PARAFAC2, no PARAFAC2 constraint is imposed on the subject-specific factors. In both cases, the recovered subject-specific profiles can differ in shape and not just in magnitude, as is the case in CP.

# 3 Experiments

## 3.1 Experimental set-up

We compare CP, PARAFAC2 and CMF with non-negativity imposed on all modes, in the task of uncovering individual time profiles. In order to choose the number of components we empirically investigate the reproducibility and replicability of the models. For reproducibility, we compare the solutions given by different initializations. For replicability, we compare the patterns uncovered by fitting the models in different stratified subsets of the input. An outline of the procedure is given in Figure 2. The complete results regarding model selection can be found in the full article [4].

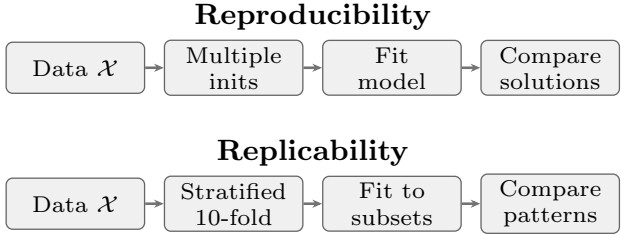

**Figure 2: Outline of the reproducibility and replicability checks utilized for model selection.**

CP is fitted using `non_negative_parafac` from TensorLy [9]. CMF and PARAFAC2 are fitted using MatCoupLy [15], which supports non-negativity constraints on all modes via alternating optimization - alternating direction method of multipliers (AO-ADMM) [16] and handles missing data via expectation maximization (EM) [5]. For each experiment, 50 random initializations were used, selecting the run with the lowest objective value. Runs exceeding the maximum iterations (15000) or with AO-ADMM feasibility gaps above $10^{-5}$ were discarded.

## 3.2 Longitudinal metabolomics data

*Dataset.* The data analyzed in this experiment constitutes a subset of nuclear magnetic resonance (NMR) spectroscopy and hormone measurements of blood samples collected during a meal challenge test from the COPSAC$_{2000}$ cohort [2] after overnight fasting. Measurements were taken at $t = 0\,hr, 0.25\,hr, 0.5\,hr, 1\,hr, 1.5\,hr, 2\,hr,$ $2.5\,hr$ and $4\,hr$ after meal intake from 299 subjects. Using only the male subjects, a tensor of 161 metabolites/hormones$\times$ 8 *time-points* $\times$ 140 *subjects* is created. More details on the challenge test as well as a CP analysis can be found in [20]. Unlike this work, however, we do not perform T0-correction (subtracting fasting state from all time-points) to preserve the temporal trajectories. The data was scaled within the metabolites mode before analysis.

*Model selection.* For all models, $R = 6$ was selected based on reproducibility and replicability analysis [1], as well as interpretability of the extracted components.

*Results.* All three models recover comparable metabolite factors; however, substantial differences can be observed in the extracted temporal profiles. By definition, CP uncovers structurally identical profiles for all subjects, since differences can only exist in terms of scaling. BMI-related group differences were observed in

four out of six components. While three of the BMI-related components are mainly dominated by lipoproteins, one component revealed trajectories related to insulin/c-peptide and glycolysis-related metabolites (Figure 3). This is a component of interest since it achieves the highest correlations with BMI-related variables [4]. While CP, CMF and PARAFAC2 models all capture a component modelling these metabolites, we observe that, in terms of temporal trajectories, only CMF and PARAFAC2 reveal structurally distinct temporal trajectories between BMI and IR groups, which CP cannot by definition.

## 3.3 Longitudinal sensitization data

*Dataset.* We also investigate the performance of methods on a sensitization dataset from the COPSAC$_{2000}$ cohort that represents allergen specific immunoglobulin E (sIgE) levels over time for milk, egg, wheat flour, peanut, birch, timothy grass, mugwort, dog, cat, mold and house dust mite (more details can be found in [17, 19]). The formed dataset is a 11 *allergens* $\times$ 6 *time-points* $\times$ 176 *subjects* tensor. Subjects with missing entries were omitted. The data was log-transformed and scaled within the allergens mode before analysis.

*Model selection.* For all models, $R = 5$ was selected based on reproducibility and replicability analysis, as well as interpretability of the extracted components. We note that CP is not as replicable as CMF and PARAFAC2.

*Results.* All models recover two components related to food allergens (milk, egg, wheat flour and peanut), two components on aeroallergens (birch, timothy grass, mugwort, dog, cat, mold and mite) and one on a mix of both. These results are in line with previous studies on this dataset [19]. Similar to the metabolomics application, the increased flexibility of CMF and PARAFAC2 enables the capture of structurally richer subject-specific temporal profiles. In particular, in one of the food-related components, CMF and PARAFAC2 reveal clear delivery-mode shape differences (natural and c-section) that CP fails to capture (Figure 4).

# 4 Conclusion and discussion

Non-negative CMF and PARAFAC2 recover individual time profiles in two distinct settings, capturing shape differences across subjects that CP cannot. In metabolomics, the proposed approaches uncover heterogeneity in subject-specific temporal profiles that can be stratified by BMI/IR, while in sensitization, observed differences that can be stratified by delivery mode. In both cases, such structure remains hidden under CP's more restrictive assumptions.

While CMF and PARAFAC2 recover comparable feature and temporal patterns, subject-level profiles can differ substantially across models (see [4]). Since CMF imposes no additional constraint other than non-negativity on the subject-specific temporal factors, it should be preferred for non-negative data; PARAFAC2 remains the better choice when negative entries are allowed.

A key limitation of both approaches is missing samples: a missing visit for subject $k$ introduces a missing column in $\mathbf{X}_k$, breaking model uniqueness. We excluded such subjects; addressing this via temporal smoothness constraints is left as future work. Sources of individual variation could be further explored by joint analysis

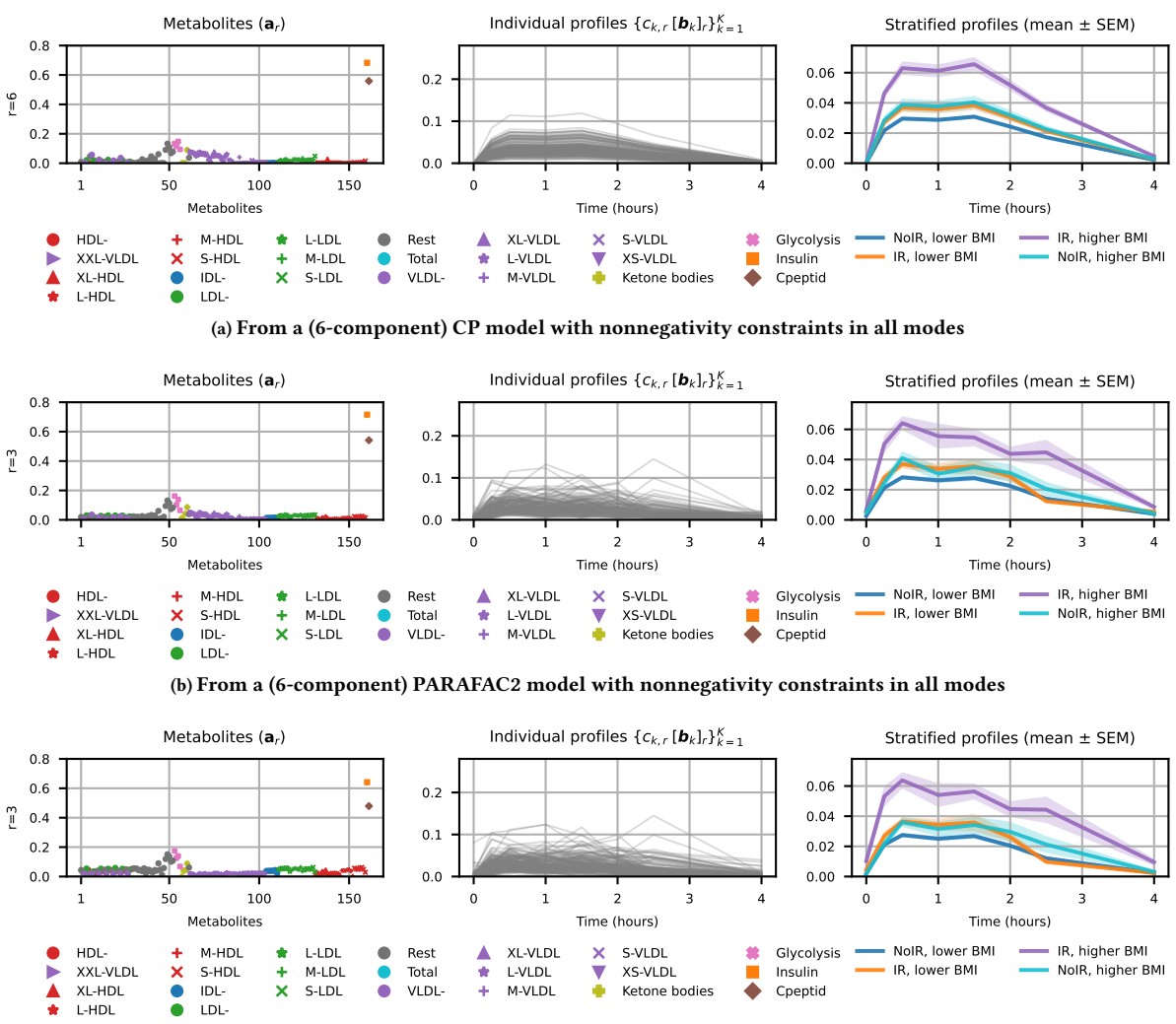

(a) From a (6-component) CP model with nonnegativity constraints in all modes

(b) From a (6-component) PARAFAC2 model with nonnegativity constraints in all modes

(c) From a (6-component) CMF model with non-negativity constraints in all modes

**Figure 3: The component modelling insulin, c-peptide and glycolysis-related metabolites extracted using different models.** $\mathbf{a}_r$ **denotes the pattern in the metabolites mode. Subject-specific time profiles scaled by the corresponding subject scores, i.e.,** $c_{k,r}[\mathbf{b}_k]_r$**, are shown in the middle column. Scaled subject-specific time profiles colored according to BMI/IR groups are shown in the last column. SEM refers to standard error of the mean.**

with other data modalities. Validation across independent cohorts also remains future work.

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
