# OpenReview forum: "Revealing Subject-Specific Temporal Trajectories From Longitudinal Data"
_KDD.org/2026/Workshop/TensorKDD — KDD 2026 Workshop TensorKDD Oral_

### Official Review · Reviewer_Q7yF · 2026-06-08
**Revealing Subject-Specific Temporal Trajectories From Longitudinal Data**

**Rating:** Accept
**Confidence:** 3
**Best Paper Recommendation:** No

**Review:**

## Summary

This paper studies how to recover subject-specific temporal trajectories from longitudinal multivariate data. The central argument is that CP decomposition provides only one shared temporal profile per component, so subject-level differences are expressed mainly through scaling, whereas CMF and PARAFAC2 allow slice-specific factor matrices and can therefore represent subject-specific temporal shape variation. The authors apply non-negative CP, PARAFAC2, and CMF to two COPSAC2000 datasets: a postprandial metabolomics tensor, and a sensitization tensor. They show that CMF and PARAFAC2 reveal group-level trajectory differences associated with BMI/insulin resistance in the metabolomics data and delivery mode in the sensitization data, whereas CP cannot represent such shape differences by construction. Model order is selected using reproducibility across multiple initializations and replicability under stratified 10-fold subsampling.

## Strengths

* **Clear motivation and appropriate model framing.**
  The paper clearly explains why CP is limited for longitudinal trajectory analysis: it estimates one shared temporal profile per component, so subjects differ mainly by scaling. CMF and PARAFAC2 are well motivated because they allow subject-specific temporal profiles and can capture shape differences across individuals.

* **Relevant and interpretable biomedical applications.**
  The two COPSAC2000 datasets—postprandial metabolomics and longitudinal sensitization—are appropriate test cases for the proposed approach. The reported findings, including BMI/insulin-resistance-related metabolic trajectories and delivery-mode-related sensitization patterns, are biologically plausible and interpretable.

* **Reasonable model-selection strategy.**
  The use of reproducibility across random initializations and replicability across stratified subsets is a sensible choice for non-convex factorization models. This is stronger than relying only on reconstruction fit, especially for tensor and coupled matrix decompositions.

## Weaknesses

* **Limited methodological novelty.**
  The paper mainly applies existing CMF and PARAFAC2 methods to new biomedical datasets rather than proposing a new model, algorithm, or theoretical contribution. Since prior PARAFAC2 applications in neuroscience, electronic health records, and microbiome analysis are already acknowledged, the novelty should be framed more modestly and precisely.

* **Insufficient quantitative evaluation.**
  The results are mostly qualitative and figure-based. The paper does not report enough quantitative evidence, such as model fit, explained variance, reproducibility/replicability scores, effect sizes, or statistical tests for group separation. This makes it difficult to assess whether the recovered subject-specific trajectories reflect robust signal rather than overfitting.

* **Comparison with CP is partly expected by construction.**
  CP cannot capture subject-specific temporal shape differences by definition. Therefore, showing that CP fails to recover such differences is not sufficient on its own. The stronger claim should be supported by evidence that CMF/PARAFAC2 recover meaningful and replicable subject-specific structure.

* **Missing-data handling may introduce bias.**
  In the sensitization analysis, subjects with missing entries are omitted. This could introduce selection bias if missingness is associated with delivery mode, sensitization severity, or other clinical variables. Since one of the main claims concerns delivery-mode differences, the paper should report the number of excluded subjects and assess whether dropout differs across groups.

---

### Official Review · Reviewer_Qeyd · 2026-06-10
**Review for "Revealing Subject-Specific Temporal Trajectories From Longitudinal Data"**

**Rating:** Accept
**Confidence:** 4
**Best Paper Recommendation:** Yes

**Review:**

## Summary:

In this work, the authors use tensor decomposition methods to analyze temporal data in metabolomics and sensitization data analysis. This work moves beyond the traditional CP decomposition that is typically used, and argues that coupled matrix factorizations (CMF) and PARAFAC2 are actually better suited to model subject-level profiles. It appears true that CP fails to accurately model subject-level profiles, and instead smooths over them, and the results seem compelling that CMF and PARAFAC2 are better for this. Furthermore, the authors provide practical guidelines for rank-selection, reproducibility, and replicability, ensuring their pipeline is robust. This work is very relevant to the TensorKDD workshop and is overall well done. However, I believe this paper would benefit from an explicit study of the sensitivity of the rank.


## Strengths:

• The 2 methods seem to generate very similar results in Figure 3 and Figure 4 which adds credibility to these findings. The CP is clearly poor at modeling the individual profiles, as the authors note.

• The authors outline their criteria for empirically selecting the rank (which includes stability across runs), which helps guide those who want to replicate or extend the study.

• The authors provide supporting literature to defend some of their results.


## Weaknesses:

• I would appreciate further analysis of the rank selection, even as simple as plotting the reconstruction error as a factor of the rank, to help gain intuition on the sensitivity of the rank.

---

### Official Review · Reviewer_1w57 · 2026-06-11

**Rating:** Accept
**Confidence:** 3
**Best Paper Recommendation:** Yes

**Review:**

This paper employs Couple Matrix Factorization (CMF) and PARAFAC2 as alternatives to CP for revealing subject-specific temporal trajectories in longitudinal multivariate data. The approach is evaluated on two COPSAC2000 datasets: postprandial metabolomics data and longitudinal allergen sensitization data. Results show that CMF and PARAFAC2 recover temporal shape differences associated with BMI/insulin resistance and delivery mode that CP fails to capture.

$\textbf{Strengths:}$
* The paper addresses a meaningful limitation of CP for longitudinal data.
* The motivation is clear and practically relevant
* The paper compares three appropriate tensor methods: CP, PARAFAC2, and CMF.

$\textbf{Weaknesses:}$
* The reproducibility and replicability steps utilized for model selection aren’t clear
* The biological or clinical significance of the discovered differences is not discussed
* It is unclear why non-negative constraint was employed. It would be good to have a single sentence explaining the reasoning to someone not familiar with the metabolomics area.

$\textbf{Things to improve:}$
* I’m not sure if it’s because I’m not used to these terms, but “planned visits in longitudinal cohorts”, “challenge tests”, and “longitudinal metabolomics” in the abstract are a little hard to get at a first glance. Also, most of the paper contains similar terms that are specific for the metabolomics area that would benefit for a single sentence explaining their meaning to CS people that aren't familiar with them.
* It is unclear how robust the extracted trajectories are to preprocessing choices such as scaling, log transformation, excluding missing subjects, and not applying T0 correction.
* The paper does not provide reconstruction errors, fit quality, or quantitative comparison across CP, PARAFAC2, and CMF.
* It would be good to briefly mention the contributions in the introduction.
* The motivation around “uncovering underlying patterns” is somewhat abstract. It would help to give concrete examples of the types of patterns the authors aim to reveal